# Desalination Characteristics of Cellulose Acetate FO Membrane Incorporated with ZIF-8 Nanoparticles

**DOI:** 10.3390/membranes12020122

**Published:** 2022-01-21

**Authors:** Tong Li, Yuhong Wang, Xinyan Wang, Caixia Cheng, Kaifeng Zhang, Jie Yang, Guangshuo Han, Zhongpeng Wang, Xiuju Wang, Liguo Wang

**Affiliations:** 1School of Water Conservancy and Environment, University of Jinan, Jinan 250022, China; L17866929054@163.com (T.L.); a15254138103@163.com (C.C.); 15621789268@163.com (K.Z.); jjjyang922@163.com (J.Y.); hangs98@163.com (G.H.); chm_wangzp@ujn.edu.cn (Z.W.); 2National Center of Ocean Standards and Metrology, Tianjin 300112, China; hywyh@163.com; 3Shandong Zhaojin Motian Co., Ltd., Zhaoyuan 265400, China; zy_wxy@163.com; 4Shandong Key Laboratory of Water Pollution Control and Resource Reuse, Shandong University, Qingdao 266237, China

**Keywords:** desalination, ZIF-8, forward osmosis membranes

## Abstract

Forward osmosis membranes have a wide range of applications in the field of water treatment. However, the application of seawater desalination is restricted, so the research of forward osmosis membranes for seawater desalination poses new challenges. In this study, zeolitic imidazolate framework-8 (ZIF-8) was synthesized by a mechanical stirring method, and its crystal structure, surface morphology, functional group characteristics, thermochemical stability, pore size distribution and specific surface area were analyzed. The cellulose acetate (CA)/ZIF-8 mixed matrix forward osmosis membrane was prepared by using the synthesized ZIF-8 as a modified additive. The effects of the additive ZIF-8 content, coagulation bath temperature, mixing temperature and heat treatment temperature on the properties of the CA/ZIF-8 forward osmosis membrane were systematically studied, and the causes were analyzed to determine the best membrane preparation parameters. The structure of the CA membrane and CA/ZIF-8 mixed matrix forward osmosis membranes prepared under the optimal conditions were characterized by Fourier Transform infrared spectroscopy (FTIR), Scanning electron microscopy (SEM), contact angle and Atomic force microscope (AFM). Finally, the properties of the HTI membrane (Membrane manufactured by Hydration Technology Innovations Inc.), CA forward osmosis membrane and CA/ZIF-8 mixed matrix forward osmosis membrane were compared under laboratory conditions. For the CA membrane, the water flux and reverse salt flux reached 48.85 L·m^−2^·h^−1^ and 3.4 g·m^−2^·h^−1^, respectively. The reverse salt flux and water flux of the CA/ZIF-8 membrane are 2.84 g·m^−2^·h^−1^ and 50.14 L·m^−2^·h^−1^, respectively. ZIF-8 has a promising application in seawater desalination.

## 1. Introduction

The scarcity and pollution of freshwater resources have aroused widespread concern around the world [1]. With the growth of the population, the improvement of living standards and the upgrading of consumption patterns, the demand for water in various industries has become increasingly prominent. Sustainable development must be achieved so that future generations can also enjoy a high quality of life, rather than bear the environmental costs [2]. Membrane separation technology, due to environmental friendliness, high separation efficiency and green sustainability, has become a viable option to alleviate the shortage of water resources [3,4]. Compared with other membrane separation methods, forward osmosis (FO) is a promising seawater desalination technology. FO has the advantages of not requiring external pressure, a low energy consumption, and a good anti-pollution performance [5,6]. Among them, the mixed matrix forward osmosis membrane is a typical membrane used in the FO process. It refers to the chemical modification of the membrane by adding modified particles to the casting solution. This method can make the additives and the matrix complement each other in order to improve the performance of the membrane through improving its hydrophilicity and anti-pollution performance as much as possible. It is believed that the modification of the membrane with porous nanoparticles can increase the permeation flux of the membrane, and, at the same time, can provide a high salt rejection through the combined effect of steric hindrance and Donnan exclusion [7]. At the same time, this method also has certain limitations, such as the agglomeration of nanoparticles under a high load and a poor membrane stability and antifouling performance. To this end, researchers have discussed the nanomaterials to be added.

In previous studies, the nanomaterials added mainly include Titanium dioxide (TiO_2_), Silicon dioxide (SiO_2_), Graphene oxide (GO), Carbon nanotubes (CNT) and so on. Wang et al. [8] added TiO_2_ to the forward osmosis membrane. In the Ca^2+^ solution, the composite membrane reduced the fouling by sodium alginate (SA). However, with the addition of TiO_2_, the nano-material composite membrane had some defects. Zhang et al. [9] modified the FO membrane with SiO_2_@Multi-walled carbon nanotubes (MWNTs). Compared with the unmodified membrane, the water flux and selectivity of the prepared CA/ZIF-8 membrane were significantly improved, with the J_S_/J_V_ value dropping from 1.10 g·L^−1^ to 0.19 g·L^−1^. However, SiO_2_ cannot interact with the hydrophobic polymer due to its high-density Si-OH group, resulting in an agglomeration of SiO_2_ on the membrane surface [10]. Zhao et al. [11] prepared a novel composite forward osmosis membrane using Graphene Oxide/Multiwall Carbon Nanotube (GO/MWCNT). The results show that GO/MWCNT composite membranes provided many water transport channels, and that the water flux was improved. However, the GO/MWCNT interlayer was thick and rough, resulting in an increased and uneven membrane thickness, which affected the transport of water molecules, and the inner concentration polarization phenomenon became obvious.

As a new type of porous nano-material, Metal organic framework (MOF) material is composed of metal ions or metal clusters as the center and is connected with an organic framework [12]. MOF has the advantages of a high specific surface area, adjustable pore size, etc., has attracted widespread attention and has been used in catalysis [13,14], adsorption [15,16], sensors [17,18], separation [19,20] and other fields. However, in practice, most MOFs are considered too fragile to be used. Fortunately, zeolitic imidazolate frameworks (ZIFs) have zeolite structures that compete with hard inorganic materials and have a high porosity. Therefore, ZIFs materials are expected to be commercialized. ZIF-8 is a kind of pore MOF zeolite material composed of a zinc ion and 2-methylimidazole ligand [21,22]. Compared with other MOF materials, ZIF-8 has a superior thermal stability, chemical stability and large specific surface area [23]. Some researchers have used ZIF-8 to prepare FO membranes for water treatment. Beh et al. [24] used ZIF-8 to prepare a FO membrane with a high chemical stability and good water permeability, which was used to remove oil emulsion in water. Fu et al. [25] developed a new type of forward osmosis membrane using a polyvinyl alcohol/polydopamine-coated zeolite imidazolate framework (PVA/PDA@ZIF-8), which improved the water flux and anti-fouling performance of the membrane. Qiu et al. [26] used polydopamine-modified ZIF-8 to prepare a forward osmosis membrane for the efficient removal of heavy metal ions. After the FO membrane was modified, the retention rate of heavy metals (Cu^2+^, and Ni^2+^ and Pb^2+^) was over 96%.

Cellulose acetate (CA) is widely used as membrane material in various membrane separation processes due to is advantages of having a high hydrophilicity, pollution resistance, appropriate chlorine resistance and low cost [27,28,29]. However, the cellulose acetate membrane has poor mechanical strength and low oxidative properties [30,31]. To overcome these problems, pure CA membranes need to be modified. The characteristics of ZIF-8 can compensate for the lack of a CA membrane. However, the application of the CA/ZIF-8 asymmetric membrane in desalination has not been found in the existing literature.

In theory, ZIF-8 can provide a faster flow channel for water molecules, achieve a high salt rejection and better compatibility with polymers. The pore size of ZIF-8 is 3.4 Å, which is between the typical salt ion (6.6–8.6 Å) [32] and the size of water molecules (2.7 Å) [33], making it an ideal water molecular sieve. Zhao et al. [34] tried to add ZIF-8 nanomaterials to thin-film nanocomposite (TFN) membranes for research on seawater desalination. It was found that when the loading of ZIF-8 reached 0.15 wt%, the water flux and salt rejection rate of the TFN membrane reached 2.61 LMH·bar^−1^ and 98.6%, respectively, which had a good desalination effect. The purpose of this work is to prepare a ZIF-8 mixed matrix forward osmosis membrane with a high water flux and excellent salt rejection effect.

## 2. Materials and Methods

### 2.1. Materials

N,N-dimethylacetamide (DMAc), 1,4-dioxane, methanol, polyethylene glycol 400 (PEG-400) were purchased from Tianjin Kemel Chemical Company (Tianjin, China). The 2-methylimidazole and zinc nitrate hexahydrate (Zn(NO_3_)_2_·6H_2_O) were purchased from McLean (Shanghai, China). Cellulose acetate (CA) was purchased from Aladdin (Shanghai, China). Sodium chloride (NaCl) was produced from Tianjin Jinbei Fine Chemical Co., LTD. (Tianjin, China), and 100-mesh polyester non-woven fabric was produced from Suzhou Wenyi Weaving Co., LTD (Suzhou, China). All of the reagents used were analytical grade. The chemical used in this study has not been further purified.

### 2.2. Preparation of MOFs and Composite Membranes

#### 2.2.1. Synthesis of ZIF-8

ZIF-8 material was prepared according to the previous report [30,35]. A total of 3.0 g zinc nitrate hexahydrate (Zn(NO_3_)_2_·6H_2_O) was added into 100 mL methanol to prepare liquid A. A total of 6.6 g 2-methylimidazole was added to 100 mL methanol to prepare liquid B. Liquid A was poured into liquid B and stirred at 25 °C for 1 h to make the solution fully mixed and reacted. After washing and centrifugation with methanol 3 times, the white precipitate was collected. After drying for 12 h in an oven at 60 °C, the material was ground and collected.

#### 2.2.2. FO Membrane Preparation

The preparation process of CA/ZIF-8 membrane is shown in Figure 1 [36]. During the preparation of the membranes, N,N-dimethylacetamide(DMAc) [37] was used as the solvent, 1,4-dioxane [38] as a cosolvent, PEG-400 [39] as a pore-making agent and ZIF-8 as additive. First, the solvent, cosolvent, pore-forming agent, and additives were added to the three-necked flask in sequence, and ultrasound was performed until the additives could be fully dissolved. Then, a certain amount of dry CA [40,41] was added to the three-necked flask, and the casting solution was heated in an oil bath at a constant temperature for 8 h. Next, after the casting solution was degassed at room temperature for 24 h, an appropriate amount was poured on the dry glass plate covered with polyester gauze, and then the membrane was scraped with a glass rod. After evaporating for 30 s at room temperature, the glass plate was placed in a coagulation bath at a certain temperature. After the membrane came off automatically, it was soaked in deionized water for later use. The prepared membranes required heat treatment for 15 min prior to testing. The compositions of the casting solutions are shown in Appendix A.

### 2.3. Characterizations

The crystal structure was characterized by X-ray powder diffraction (XRD, Bruker D8 advance). The Nicolet iS50 FTIR manufactured by Thermo Scientific was used to analyze the functional groups of samples. The thermal stability of the synthesized ZIF-8 under N_2_ atmosphere was recorded using TGA55 thermal analyzer. The TG curve reflects the initial decomposition temperature, the termination temperature of the substance and the mass of the remaining material. DTG reflects the maximum weight loss temperature of the ZIF-8. The crystal morphology was studied by SEM of QUANTA 250 FEG. The surface and section morphology of the membrane were also observed by SEM. The N_2_ adsorption/desorption isotherms of the synthesized ZIF-8 materials were studied by the BET analysis method, and the pore size distribution and specific surface area were obtained. Contact angles were obtained at 25 °C using a contact angle tester (Germany Dataphysics OCA40) to evaluate the surface hydrophilicity of the membrane. The surface roughness of the membranes was analyzed by AFM. The membrane needs to be dried before testing with AFM.

### 2.4. Performance Testing of FO Membrane Characterizations

In the laboratory at room temperature, 1M NaCl was used as the draw solution and deionized water was used as the raw material to test the forward osmosis performance of the membrane. Through the peristaltic pump (WT600-2J, rotating speed 100 rpm), the flow rates of the raw material liquid and traction liquid on both sides of the membrane were adjusted to be the same, using a cross-flow method. A beaker containing NaCl solution was placed on one side of the balance, and the water flux was obtained by measuring the change in the quality of the NaCl solution. At the same time, the deionized water was put on the side of the conductivity meter and the change in NaCl concentration in the original solution was measured to calculate the reverse salt flux of the membrane. The indicators of the electronic balance and conductivity meter were recorded every 1 min, and the experiment was stopped after recording for 1 h. A total of 60 data points were obtained.

The separation performance of FO membranes can be measured by the reverse salt flux and water flux. The reverse salt flux reflects the salt interception effect of the membrane, and the water flux reflects the water permeability of the membrane.

The water flux (*J_w_*) was obtained by calculating the increase in the mass of the suction side over a period of time, as shown as Equation (1), and its unit is L·m^−2^·h^−1^.
(1)$$J_{W}=\frac{\Delta m}{\rho \times S\times t}$$
where ∆*m* is the mass increase in the extraction solution side, and the unit is kg; *ρ* represents the density of water, and its unit is kg·m^−3^; the effective area of the membranes is represented by *S*, and the unit is m^2^; *t* is the running time of the system, and the unit is h. An electronic balance (AR4202CN) was used to monitor the quality change in the feed liquid.

The reverse salt flux was calculated as follows, as shown as Equation (2), and its unit is g·m^−2^·h^−1^.
(2)$$J_{S}=\frac{C_{t}V_{t}-C_{0}V_{0}}{S\times t}$$

*C*_0_ (mol/L) and *V*_0_ (L) represent the initial salt content and initial volume of the raw material solution, respectively; *C*t (mol/L) and *V*t (L) represent the salt content and volume of the peristaltic pump after running for *t* (h) time, respectively; the effective area of membranes is represented by *S*, and its unit is m^2^. A conductivity meter (DDSJ-308) measured the conductivity to determine the salt concentration, and then a standard concentration–conductivity curve was followed.

## 3. Results

### 3.1. Characterisation of ZIF-8

XRD spectra are mainly used for qualitative phase analysis. In Figure 2, the spectra of the synthesized ZIF-8 are compared with the theoretical spectra. The X-ray diffraction pattern of the synthesized ZIF-8 material is consistent with the theoretical simulation results in the literature. The characteristic peaks appeared at the two theta values of 7.4°, 10.4°, 12.8°, 14.7° and 18.0°, corresponding to (011), (002), (112), (022) and (222) crystal planes, respectively, which were consistent with the previous reports [31,42]. This shows that the ZIF-8 crystal was successfully prepared in the experiment. In addition, according to the peak intensity in the X-ray spectrum, the synthesized ZIF-8 material has a high crystallinity.

Figure 3 represents the FTIR spectra of the synthesized ZIF-8. In the structure of ZIF-8, the weak tensile vibrations at 3134 cm^−1^ and 2925 cm^−1^ represent the aromatic and aliphatic C-H bonds, respectively. The strong weak vibration near 2933 cm^−1^ is caused by the ring stretching of imidazole groups. The peak value at 1587 cm^−1^ is related to the stretching vibration of C-N bond. The value at 754 cm^−1^ corresponds to the Zn-O bond, and the value at 1147 cm^−1^ corresponds to the C-N bond. The stretching vibration peak of Zn-N is at 422 cm^−1^.

The morphology of ZIF-8 was analyzed by SEM. It can be seen from Figure 4 that the synthesized ZIF-8 material in the experiment has a certain agglomeration phenomenon. The prepared ZIF-8 nanoparticles exhibit a typical regular polyhedron structure, with a smooth surface and good uniformity, which can be used to prepare high-quality mixed matrix membranes. The structure of this nanomaterial has been reported in the literature and can be identified as ZIF-8 nanoparticles [43].

Figure 5 analyzes the thermal stability of the synthesized ZIF-8 in the temperature range from room temperature to 800 °C. In the range of room temperature to 500 °C, the material of ZIF-8 has a slight weight loss, which is probably caused by the dehydration of the skeleton in ZIF-8. When the temperature reaches approximately 600 °C, ZIF-8 appears to have an obvious weight loss phenomenon, and the rate of mass decline is obviously accelerated. This is due to the decomposition of ZIF-8 particles. After the complete decomposition of ZIF-8, the solid remains of ZnO are left. ZIF-8 nanoparticles exhibit an excellent thermal stability, which is attributed to the presence of coordination bonds [44]. Through analysis, it can be concluded that the thermal stability temperature of synthesized ZIF-8 in the experiment is 600 °C, which is consistent with previous reports in the literature [32,45].

The pore size distribution and specific surface area of synthesized ZIF-8 were analyzed by measuring the N_2_ adsorption and desorption isotherms. The adsorption isotherms belong to type I [46,47], and the pore size distribution of synthesized ZIF-8 mainly concentrates on 2–30 nm, which is mesoporous [48]. There is a small hysteresis loop due to the capillary condensation of the sample. The mesoporous structure of ZIF-8 is attributed to the accumulation of smaller nanocrystals [49]. As nanoparticles pile up, large pores can be formed. The BET method was used to calculate the pore size distribution and pore structure parameters of samples, as shown in Figure 6 and Table 1. The Brunauer-Emmett-Teller (BET) specific surface area, average pore size and pore volume of ZIF-8 reached 1338.67 m^2^·g^−1^, 4.30 nm and 0.82 cm^3^·g^−1^, respectively.

### 3.2. Optimization of CA/ZIF-8 FO Membranes

The effects of the ZIF-8 content, coagulation bath temperature, mixing temperature and heat treatment temperature on the properties of the CA/ZIF-8 mixed matrix forward osmosis membrane were discussed. The test conditions are shown in Appendix A.

When the content of ZIF-8 exceeds 1.0 wt%, most of the ZIF-8 nanoparticles cannot be dispersed in organic solvents, which may affect the viscosity of the casting solution [50] and make it difficult to form membranes. Therefore, the variation trends of the water flux and reverse salt flux of the CA/ZIF-8 membrane with a ZIF-8 content of 0.2 wt%, 0.4 wt%, 0.6 wt%, 0.8 wt% and 1.0 wt% were selected to be discussed. As can be seen from Figure 7a, when the material content of ZIF-8 increased from 0.2 wt% to 0.6 wt%, both the water flux and reverse salt flux showed an upward trend. This is because the incorporation of ZIF-8 material improves the hydrophilicity of the forward osmosis membrane, while the nanochannels based on ZIF-8 nanoparticles create more membrane channels and promote water permeability [51,52]. When the content of ZIF-8 ranged from 0.6 wt% to 1.0 wt%, both the reverse salt flux and water flux of the CA/ZIF-8 forward osmosis membrane showed a decreasing trend. This is because the content of ZIF-8 is too high to be fully dispersed in the organic solution, resulting in defects in the membranes [53], and some membrane pores are blocked, which increases the membrane resistance and reduces the water flux [54]. Moreover, when the incorporation of ZIF-8 exceeded 0.6 wt%, the viscosity of the casting solution increased, and the thickness of the membrane increased, thereby reducing the water flux [50,55,56,57]. Considering the combined water flux, reverse salt flux and additive dosage, 0.4 wt% was selected as the best content of ZIF-8.

Different coagulation bath temperatures produce a different inner structure of the CA/ZIF-8 membrane. Therefore, the appropriate coagulation bath temperature is very important to the performance of the membrane. As can be seen from Figure 7b, as the temperature of the coagulation bath increased, the water flux showed a trend of rising first and then decreasing, whereas the reverse salt flux increased first and then stabilized. When the coagulation bath temperature reached 35 °C, both the reverse salt flux and water flux reached the maximum value, which were 2.60 g·m^−2^·h^−1^ and 48.67 L·m^−2^·h^−1^, respectively. The above experimental phenomena can be explained as follows: according to the nucleation and growth mechanism, the affinity between DMAc and water is strong enough at a higher coagulation bath temperature, and the outer diffusion rate of solvent DMAc is much higher than the inner diffusion rate of the non-solvent, so the cortex is very dense, which reduces the diffusion rate of the non-solvent [58]. This facilitates the creation of finger-like structures and provides channels for water mass transfer. However, the high temperature of the coagulation bath leads to the rapid movement of the polymer chain, which makes the membrane pore shrink [59] and leads to a decrease in the water flux. At the same time, too high a coagulation bath temperature accelerates the precipitation rate of the polymer in the process of membrane formation, thickens the membrane layer [60] and leads to a decrease in membrane permeability, reducing the permeability of the membrane. After comprehensive consideration, 35 °C was selected as the optimum coagulation bath temperature for preparing the CA/ZIF-8 mixed matrix forward osmosis membrane.

The mixing temperature has a great influence on the microstructure of the membrane. It can be seen from Figure 7c that, with an increase in the mixing temperature, both the water flux and reverse salt flux showed a trend of first increasing and then decreasing. Increasing the mixing temperature of the casting solution improves the water flux and salt cutting rate, which can be attributed to the decrease in the viscosity of the casting solution. With an increase in the mixing temperature, the curing rate of the membrane accelerates, the pore size becomes larger and the supporting layer inside the membrane becomes loose, resulting in an increase in the water flux. When the mixing temperature was above 60 °C, both the reverse salt flux and water flux of the CA/ZIF-8 mixed matrix membrane decreased. This may be due to the high mixing temperature, which changes the structure of CA, leading to the pore size contraction of the CA/ZIF-8 mixed matrix forward osmosis membrane [61]. In the case of a high temperature, polymer chain rearrangement is also very serious, and active macromolecular chains gather and aggregate on the membrane surface to form a dense cortex, resulting in a decrease in the reverse salt flux and water flux. By comparing the reverse salt flux and water flux of the forward osmosis membrane of the CA/ZIF-8 mixed matrix at mixing temperatures of 50 °C and 60 °C, it can be found that the change in the water flux was not obvious at these two temperatures, but the reverse salt flux increased more at 60 °C. Considering the reverse salt flux, water flux and energy-saving effect, the mixing temperature of 50 °C was selected as the best preparation condition of the CA/ZIF-8 mixed matrix forward osmosis membrane.

The effect of the heat treatment temperature on the CA/ZIF-8 membrane was discussed. The experimental results are shown in Figure 7d. As the heat treatment temperature increased, the water flux first increased and then decreased, reaching the highest value of 50.02 L·m^−2^·h^−1^ at 60 °C. The reverse salt flux also showed the same trend, and the maximum value was 2.60 g·m^−2^·h^−1^. With the increase in the heat treatment temperature, the hydrophilicity of the ZIF-8 material was greatly enhanced, and the traction force of the inner pores to water was also increased accordingly [62,63]. When water molecules come into contact with the membrane, water molecules can be transferred from the inside of the membrane pores, thereby increasing its water flux. In addition, the increase in the heat treatment promoted the release of the residual solvent, and the pore size of the membrane increased, which also led to an increase in the water flux and a decrease in the salt removal rate [36]. When the heat treatment temperature was higher than 60 °C, both the water flux and reverse salt flux decreased. This phenomenon can be explained as due to the excessive heat treatment temperature making the membrane “dehydrate and shrink”, and the pore diameter of the membrane also shrinking, leading to a decrease in the water flux and an increase in the salt cutting rate [64]. From a further comparison of the water flux and reverse salt flux at 50 °C and 60 °C, it is found that the water flux at these two temperatures did not change significantly, whereas the reverse salt flux was smaller at 50 °C. Considering the water flux, reverse salt flux and energy-saving benefit, 50 °C was selected as the optimal heat treatment temperature for preparing the CA/ZIF-8 mixed matrix forward osmosis membrane.

### 3.3. Characterization of FO Membranes

Through the above experimental study, the optimal preparation parameters of the CA/ZIF-8 mixed matrix forward osmosis membrane were obtained: the optimal content of ZIF-8 particles was 0.4 wt%, the coagulation bath temperature was 35 °C, the mixing temperature was 50 °C and the heat treatment temperature was 50 °C. The structures of the CA/ZIF-8 mixed matrix forward osmosis membrane and CA forward osmosis membrane prepared under the optimum conditions were characterized.

#### 3.3.1. Water Contact Angle of the Prepared Membranes

The contact angle reflects the hydrophilicity of the membrane. Generally speaking, the smaller the contact angle, the better the hydrophilicity of the membrane. Figure 8 shows an analysis of the water contact angle test results of the CA membrane and CA/ZIF-8 membrane. As far as we know, if hydrophobic ZIF-8 nanoparticles are exposed to the membrane surface, they exhibit an increase in the water contact angle. However, the introduction of ZIF-8 makes the contact angle of the CA/ZIF-8 membrane decrease slightly, and the contact angle value was 66.6°. The water contact angle of the CA membrane without ZIF-8 modification was 69.7°. This is because the addition of ZIF-8 nanoparticles reduces the cross-linking degree of the membrane surface structure, and a lower cross-linking degree means a higher water permeability.

#### 3.3.2. Morphology of the Prepared Membranes

The prepared CA forward osmosis membrane and CA/ZIF-8 mixed matrix forward osmosis membrane were tested by a scanning electron microscope to observe the surface and cross-sectional morphology of the membrane. In Figure 9(a_1_,b_1_), the epidermis of the CA membrane and the CA/ZIF-8 mixed matrix membrane were relatively dense. In addition, compared with the CA forward osmosis membrane, the deposition of ZIF-8 nanoparticles on the surface of the CA/ZIF-8 mixed matrix membrane was very small. Compared with the cross-section (a_2_) and (b_2_) of the forward osmosis membranes, it was found that the finger-like pore structure of CA/ZIF-8 mixed matrix FO membranes was more obvious, the number of pores increased and the arrangement was more compact. This phenomenon can improve the water flux and salt removal rate of the FO membrane, and can thus slow down the internal concentration polarization (ICP) of the FO membrane.

#### 3.3.3. AFM of the Prepared Membranes

AFM was used to analyze the surface morphology and roughness of the CA forward osmosis membrane and CA/ZIF-8 forward osmosis membrane, and the brighter region represents the higher surface roughness of the membrane. Specific images and roughness parameters are shown in Figure 10. Roughness parameters refer to the root mean square roughness (Rq) and average roughness (Ra). It can be observed that the addition of ZIF-8 particles increases the average roughness Ra of the CA/ZIF-8 matrix forward osmosis membrane from 4.58 nm to 5.49 nm, and the root mean square roughness Rq from 5.93 nm to 6.87 nm. This may be attributed to the increase in the surface roughness of the membrane due to the deposition of ZIF-8 nanoparticles. In general, the increase in the membrane surface roughness is beneficial to increasing the surface area of the membrane, promoting the transport of water molecules and, thus, improving the permeability of the membrane [65].

#### 3.3.4. FTIR Spectra of the Prepared Membranes

The functional groups of CA forward osmosis membranes and CA/ZIF-8 mixed matrix forward osmosis membranes were investigated by FTIR. This can be observed from Figure 11. The value at 2940 cm^−1^ is the stretching vibration of C-H; 1737 cm^−1^ is consistent with the stretching vibration of C=O; 1367 cm^−1^ and 1431 cm^−1^ represent the bending vibration of C-H; 1031 cm^−1^ and 1121 cm^−1^ correspond to the symmetric and asymmetric vibration of C-O, respectively; 900 cm^−1^ is consistent with the bending vibration peak of C-H; and the C-O group is located at 1230 cm^−1^. Due to the small amount of ZIF-8 added, the corresponding peak was weak and not observed.

### 3.4. Comparison of FO Membrane Separation Performance

The properties of the CA membrane and CA/ZIF-8 mixed matrix forward osmosis membrane were tested, and the results are shown in Figure 12. During the test, DI was the raw material solution and the 1 M NaCl solution was the extraction solution. The water flux of the commercial HTI membrane was 8.5 L·m^−2^·h^−1^, and the reverse salt flux was 0.79 g·m^−2^·h^−1^. The reverse salt flux and water flux of the CA membrane were 3.4 g·m^−2^·h^−1^ and 48.85 L·m^−2^·h^−1^, respectively. The reverse salt flux and water flux of the CA/ZIF-8 mixed matrix forward osmosis membrane reached 2.84 g·m^−2^·h^−1^ and 50.14 L·m^−2^·h^−1^, respectively. It can be known from the experimental results that the water flux of the CA/ZIF-8 mixed matrix forward osmosis membrane was slightly increased compared with that of the CA membrane, and the separation effect was improved. This is due to the addition of ZIF-8, which improves the pore and roughness of the membrane.

## 4. Conclusions

The ZIF-8 material was synthesized by the mechanical stirring method, and the prepared ZIF-8 material was characterized by XRD, SEM, FTIR, thermogravimetric analysis and N_2_ adsorption and desorption analysis methods. ZIF-8 was added to the CA matrix, and the CA/ZIF-8 mixed matrix forward osmosis membrane was prepared by the phase inversion method. The ZIF-8 content, coagulation bath temperature, heat treatment temperature and mixing temperature of the CA/ZIF-8 mixed matrix forward osmosis membrane were discussed, and the optimal preparation process parameters of the membrane were obtained: the optimal content of ZIF-8 was 0.4 wt%, the coagulation bath temperature was 35 °C, the heat treatment temperature was 50 °C and the mixing temperature was 50 °C. Structure characterization and a performance analysis of the CA/ZIF-8 mixed matrix forward osmosis membrane prepared under optimal conditions were performed. It was found that the incorporation of ZIF-8 increased the surface roughness of the membrane, which was beneficial to increasing the surface area of the membrane, promoting the transmission of water molecules and, thereby, increasing the permeability of membranes. The finger-like pore structure of the mixed matrix membrane with ZIF-8 was more obvious than that of the CA membrane, and the number of pores increased, which was beneficial to improving the water flux and salt cutting rate of the FO membrane. The decrease in the contact angle of the membrane modified by ZIF-8 may be attributed to the decrease in the cross-linking degree of the membrane, which enhanced the permeability of the membrane.. The water flux of the commercial HTI membrane was 8.5 L·m^−2^·h^−1^, and the reverse salt flux was 0.79 g·m^−2^·h^−1^. The reverse salt flux and water flux of the CA membrane were 3.4 g·m^−2^·h^−1^ and 48.85 L·m^−2^·h^−1^, respectively. The reverse salt flux and water flux of the CA/ZIF-8 mixed matrix forward osmosis membrane reached 2.84 g·m^−2^·h^−1^ and 50.14 L·m^−2^·h^−1^, respectively. Compared with the CA membrane, the CA/ZIF-8 mixed matrix forward osmosis membrane has a slightly increased water flux and improved separation effect.

## Figures and Tables

**Figure 1 membranes-12-00122-f001:**
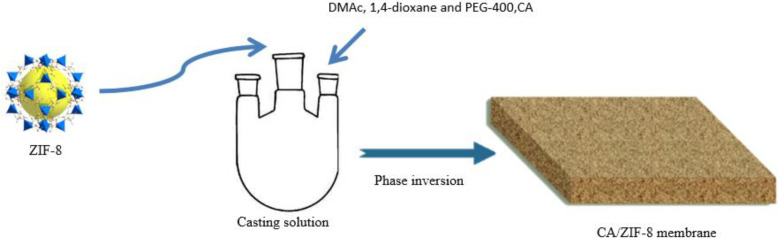
CA/ZIF-8 membrane preparation process.

**Figure 2 membranes-12-00122-f002:**
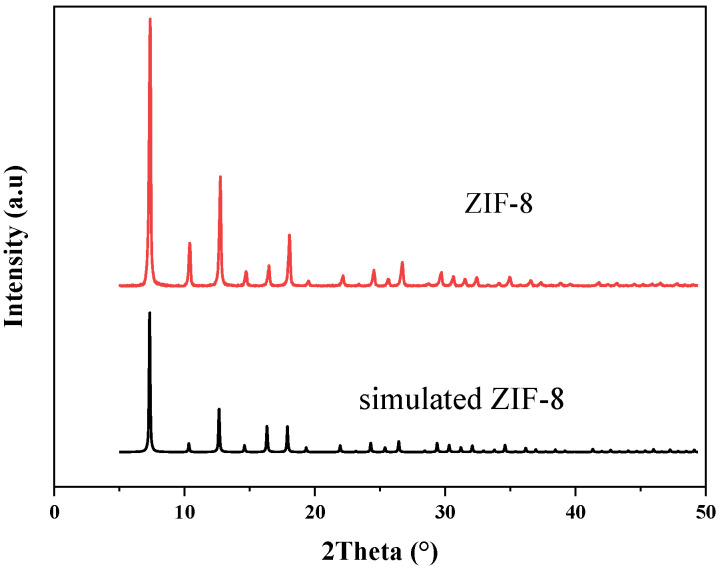
XRD spectra of ZIF-8.

**Figure 3 membranes-12-00122-f003:**
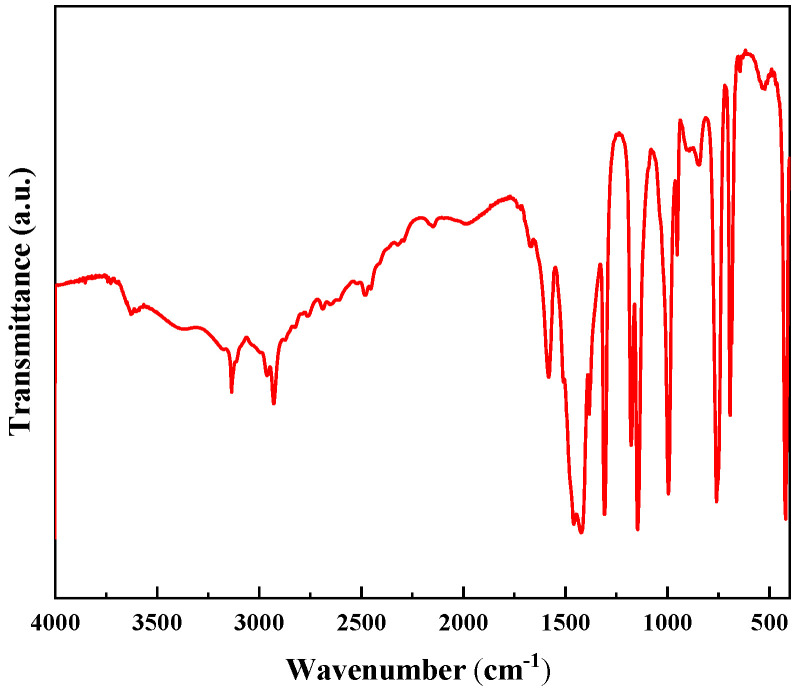
The FTIR spectra related to ZIF-8 nanocomposites.

**Figure 4 membranes-12-00122-f004:**
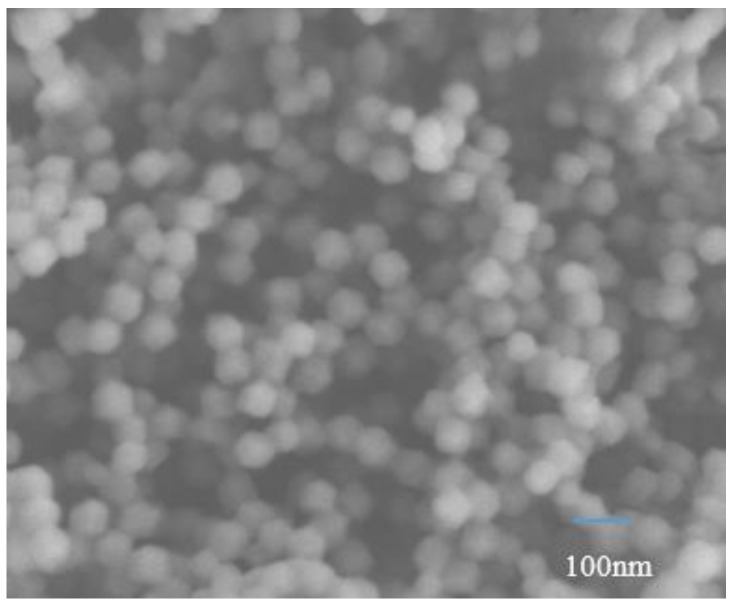
The SEM images of ZIF-8.

**Figure 5 membranes-12-00122-f005:**
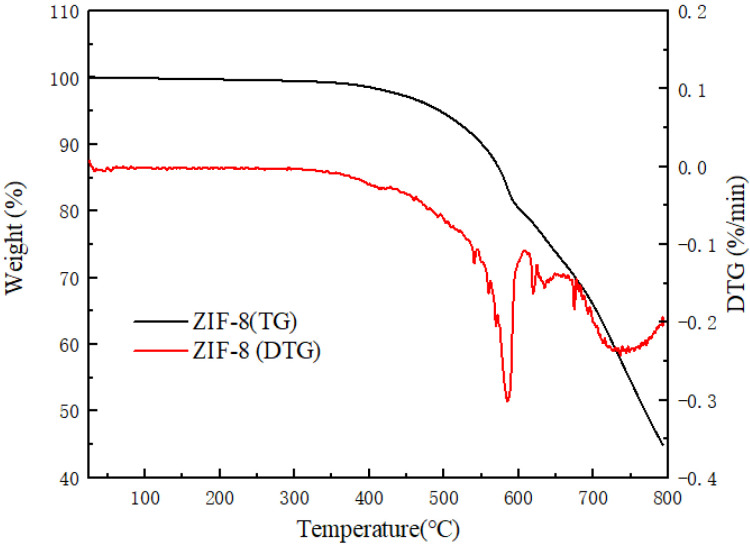
The TG and DTG of the ZIF-8 nanocomposites.

**Figure 6 membranes-12-00122-f006:**
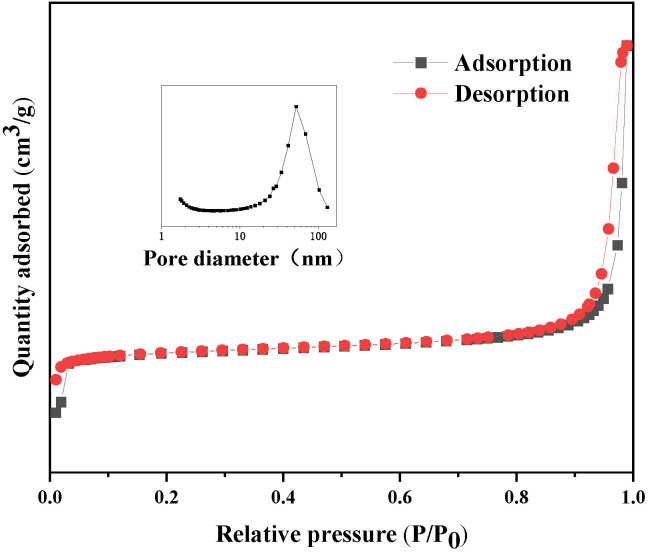
Nitrogen adsorption and desorption isotherms of ZIF-8.

**Figure 7 membranes-12-00122-f007:**
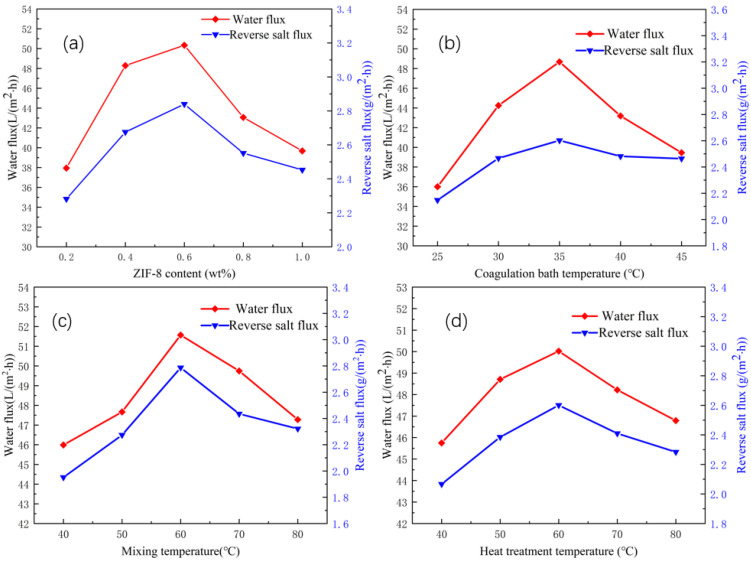
Influence of ZIF-8 content (**a**); coagulation bath temperature: 35 °C, mixing temperature: 60 °C, heat treatment temperature: 60 °C, coagulation bath temperature (**b**); ZIF-8 content: 0.4 wt%, mixing temperature: 60 °C, heat temperature: 60 °C mixing temperature (**c**); ZIF-8 content: 0.4 wt%, mixing temperature: 60 °C, heat treatment temperature: 60 °C and heat treatment temperature (**d**); ZIF-8 content: 0.4 wt%, mixing temperature: 60 °C, coagulation bath temperature: 35 °C on CA/ZIF-8 membrane.

**Figure 8 membranes-12-00122-f008:**
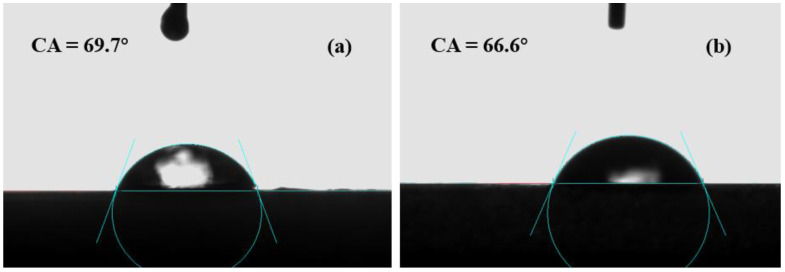
Contact angle of CA membrane (**a**) and CA/ZIF-8 membrane (**b**).

**Figure 9 membranes-12-00122-f009:**
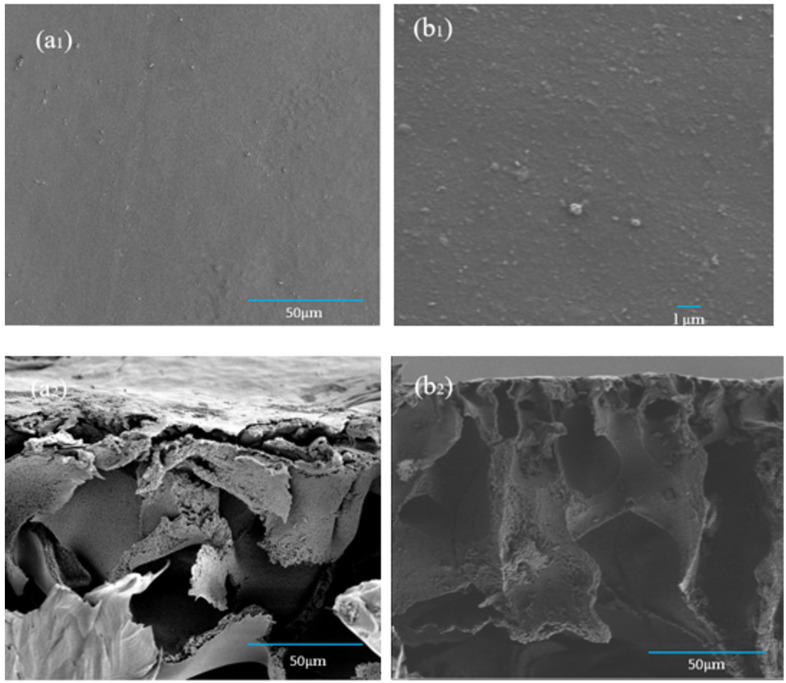
SEM images of the morphology CA membrane of surface (**a_1_**)and section (**a_2_**) and CA/ZIF-8 membrane of surface (**b_1_**)and section (**b_2_**).

**Figure 10 membranes-12-00122-f010:**
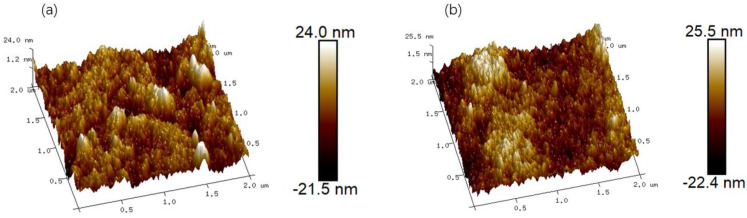
The surface AFM 3D images of the CA membrane (**a**) and CA/ZIF-8 membrane (**b**).

**Figure 11 membranes-12-00122-f011:**
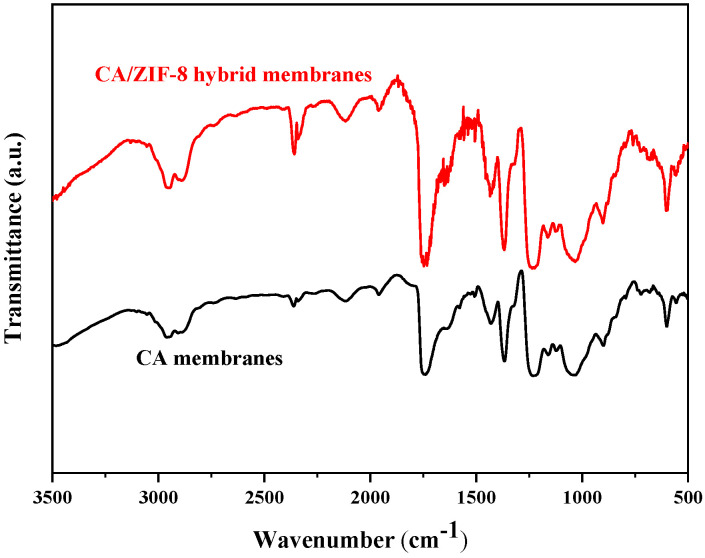
FTIR spectroscopy of CA membrane and CA/ZIF-8 membrane.

**Figure 12 membranes-12-00122-f012:**
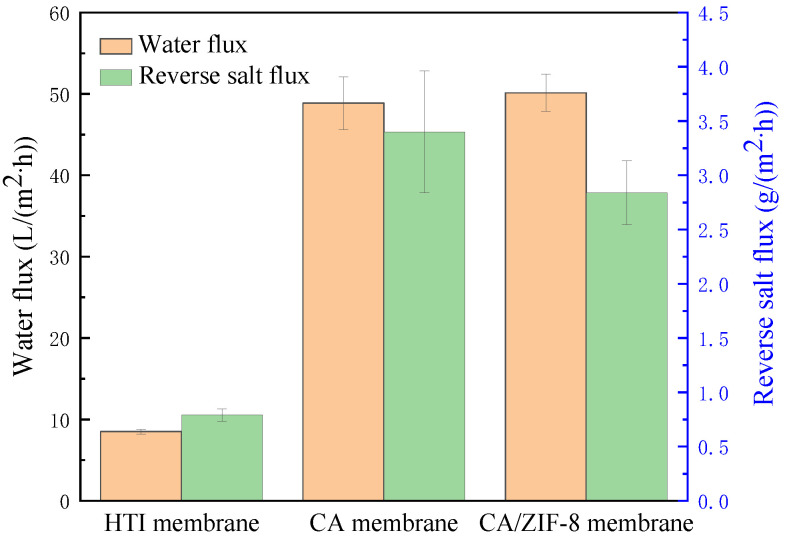
The property of the CA membrane and CA/ZIF-8 membrane.

**Table 1 membranes-12-00122-t001:** Surface area and porous structure of the ZIF-8.

Parameter	Numerical
BET Surface Area (m^2^·g^−1^)	1338.67
Langmuir Surface Area (m^2^·g^−1^)	1991.38
Micropore Area (m^2^·g^−1^)	1217.76
Pore Volume (cm^3^·g^−1^)	0.82
Average Pore Diameter (nm)	4.30

## Data Availability

Not applicable.

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
