# Peer review of "Desalination Characteristics of Cellulose Acetate FO Membrane Incorporated with ZIF-8 Nanoparticles"

_membranes, 2022, doi:10.3390/membranes12020122_

Round 1

Reviewer 1 Report

The paper entitled "Desalination characteristics of cellulose acetate FO membrane" can be accepted for publishing after accomplishing minor comments as follows:

Add abbreviation table.

Page 3, Line 105: Represent the membrane preparation in graphical form

Figure 5: No explanation of the sub-plot pore diameter in the text

Figure 8: The Figure quality should be improved. Magnification, pressure, etc., are not visible.

Page 11:  What are Ra and Rq, define separately.

Page 12: What is HTI membrane? This term is not explained anywhere throughout the manuscript.

Conclusion: It's better if the authors include some membrane characterization results in the conclusion also.

Reviewer 2 Report

This manuscript describes the preparation of a ZIF-8/cellulose acetate mixed matrix forward osmosis (FO) membrane. The ZIF-8 nanomaterial was first synthesized by mechanical stirring method prior to incorporation in the FO membrane. Several membrane preparation parameters (ZIF-8 content, coagulation bath temperature, mixing temperature, heat treatment temperature) were investigated. The biggest issue of this manuscript is the lack of additional knowledge that the methodology and the results obtained from the work could add to existing literature. The incorporation of ZIF-8, as well as the effects of the various membrane preparation parameters have all been investigated thoroughly in the past, and the authors have not even provided a sufficient literature review. Furthermore, the authors failed to highlight the novelty and importance of this work, and what sets it apart from other mixed matrix FO membrane preparation studies. The manuscript also has several language issues, that I recommend the manuscript to undergo a professional English editing service prior to resubmission. I therefore recommend that this manuscript be accepted for publication in Membranes after major revisions.

Specific comments:

  1. Improve literature review. There was no overarching story that was followed. It would be better to focus on mixed matrix membranes for FO, what previous materials were used, and why ZIF-8 was chosen (already presented). There should be a more extensive literature review of ZIF-8-incorporated FO membranes.
  2. After providing a more extensive literature review, the authors should then introduce the novelty of this work. ZIF-8 synthesis via mechanical stirring is not novel, as it was already previously reported. ZIF-8 incorporation has also been done previously. Why is this work needed?
  3. ZIF-8 characterization: NMR characterization of the product could be necessary, and compare with the structure of ZIF-8.
  4. References for the independent factors used during membrane preparation (solvent system, pore-former, solution preparation, etc.) should be cited.
  5. 3: Not clear; no scale bar
  6. Where are the details of the membrane preparation (i.e., CA concentration, ZIF loading)? All the tested parameters should also be mentioned.
  7. Were replicate experiments performed. All FO performance data do not have error bars.
  8. L232: Was solution viscosity mainly influenced by ZIF-8 content? Isn’t it more likely because of CA polymer?
  9. 8: Not clear; scale bar could not be seen
  10. Overall level of discussion should be improved.

Reviewer 3 Report

Dear Editor

Thank you for the invitation to review the manuscript entitles Desalination characteristics of cellulose acetate FO membrane incorporated with ZIF-8 nanoparticles". The manuscript described the preparation of ZIF-8 nanomaterial and uses as a nano-additive to prepare the CA/ZIF-8 mixed matrix forward osmosis membrane. The effects of additive ZIF-8 content, coagulation bath temperature, mixing temperature and heat treatment temperature on the properties of CA/ZIF-8 forward osmosis membrane which in turn on the FO performance for desalination application were studied. In this work the authors did not perform any important characterization test on all of the nanocomposite membranes such as FESEM, FTIR, EDX, thickness and contact angle etc., except the neat and the best content of the ZIF-8.

I think that this study is useful for the field of saline water desalination and therefore the manuscript can be published in Membranes after considering the following major comments.

Best Regards

Comments to the Authors:

  1. Abstract: The authors should mention the application of the FO membrane in the first sentences of the abstract not in the end because the aim of the study is to modified the membrane for saline water desalination.
  2. Abstract: there is no any information about the composition of CA and ZIF-8 in the casting solution.
  3. Abstract, line 22-23, what is HTI membrane, the authors should be define HTI.
  4. Last sentences in the introduction section line 88, please define TFN membrane.
  5. There is missing characterizations in section 2.3 such as TGA and DTG.
  6. More discussion on Figure 3 should be added.
  7. The operating conditions and parameters should be presented in the Caption of Figure 6. For example in coagulation bath temperature effect which content of nanoparticles was used?
  8. In page 7, line 232, "When the content of ZIF-8 exceeds 1.0 wt%, most of the ZIF-8 nanoparticles cannot be dissolved in organic solvents.." is the NPs dissolved in organic solvent or dispersed?
  9. In page 7, line 237, "This is because the incorporation of ZIF-8 material improves the hydrophilicity of the forward osmosis membrane, while the nanochannels based on ZIF-8 nanoparticles create more membrane channels and promote water permeability…" How the authors know that the hydrophilicity of the membrane improved with ZIF-8, there is no contact angle measurement for all composite membranes (except for neat and 0.4% ZIF-8). The discussion of the effect of ZIF-8 content should be according to the results of the characterizations performed in the experimental work.
  10. Page 7, line 241, the authors stated that "When the content of ZIF-8 ranged from 0.6 wt% to 1.0 wt%, both reverse salt flux and water flux of CA/ZIF-8 forward osmosis membrane showed a decreasing trend. This is because the content of ZIF-8 is too high, so that it cannot be fully dissolved in the organic solution, leading to the blockage of some membrane pores." If the NPs does not fully dissolved in the organic solvent the FTIR results can inform us about this phenomenon but the authors did not perform this test for all of the composite membranes. Here the authors can also confirm this phenomenon from FESEM images of the composite membrane which it did not performed for all membranes in this study except for neat and 0.4% of ZIF-8. Therefore the authors must be adding the FESEM images of the prepared membranes.
  11. Page 7, line 243, Moreover, when the incorporation of ZIF-8 exceeded 0.6 wt%, the viscosity of the casting solution increased, and the thickness of the membrane increased, thereby reducing the water flux. The authors did not measure the thickness of the membranes, sometime the contact angle overcome the effect of the membrane thickness on the performance of the membrane, therefore, the membrane thickness, and contact angle of the membranes measurement should be presented in this paper. Moreover, the morphological structure of the modified membrane has a significant effect on the performance of the membrane and it can be useful to confirm the presented results.
  12.  Page 7 line 260, This facilitates the creation of finger-like structures and provides channels for water mass transfer. However, the high temperature of the coagulation bath leads to the rapid movement of the polymer chain, which makes the membrane pore shrink and leads to the decrease of water flux. etc. How the authors discuss this results without FESEM images!!!
  13. Page 8, line 273, Similar to previous comment on The following discussion:

"With the increase of mixing temperature, the curing rate of the membrane accelerates, the pore size becomes larger, and the supporting layer inside the membrane becomes loose, resulting in the increase of water flux …"

  1. Page 8 line 296, according to the following discussion the authors should measure the pore size of all membranes in order to see the effect of the heat treatment on the membrane pore size:

 "In addition, the increase of heat treatment promoted the release of residual solvent and the pore size of the membrane increased, which also led to the increase of water flux and the decrease of salt removal rate."

  1. In order to discuss the results of FO performance, section 3.3 and its contents should be move before section 3.2.
  2. The conclusions should be revise according to the above comments.

Reviewer 4 Report

This manuscript was about the application of Desalination process and characteristics of cellulose acetate FO membrane incorporated with ZIF-8 nanoparticles

It is better to mention the abbreviation of words before using it in the first sentence

Writing the unit g/(m2·h) can be replaced with g.m-2.h-1

There’s no word ‘cellulose acetate’ in the abstract and it doesn’t need to put that word as a keyword.

It is better to re-produce figure 6 become one picture, so that, it won’t break up due to the length of page.

In line 231, I think the mixture between CA/ZIF-8 wasn’t completely dissolved through organic solvent, however, it’d produce phase separation. Please approve by adding more data and figures.

Round 2

Reviewer 2 Report

All the comments have been addressed and all the necessary revisions have been made.

Reviewer 3 Report

Dear Editor

I would like to inform you that the authors answer all my comments and therefore I accept the manuscript for publication in Membranes

Best Regards